# Hypoxia and Ezrin Expression in Primary Melanoma Have High Prognostic Relevance

**DOI:** 10.3390/ijms231810745

**Published:** 2022-09-15

**Authors:** Umberto Maccio, Alanna Mihic, Daniela Lenggenhager, Isabel Kolm, Christiane Mittmann, Mathias Heikenwälder, Anna Lorentzen, Daniela Mihic-Probst

**Affiliations:** 1Institute of Pathology and Molecular Pathology, University Hospital of Zurich, 8091 Zurich, Switzerland; 2Bavarian State Office for Health and Food Safety, 85764 Oberschleißheim, Germany; 3Department of Dermatology, University Hospital of Zurich, 8091 Zurich, Switzerland; 4German Cancer Research Center, Division Chronic Inflammation and Cancer, 69120 Heidelberg, Germany; 5Institute of Biomedicine, Institute of Molecular Biology and Genetics, Aarhus University, 8000 Aarhus, Denmark

**Keywords:** melanoma, metastasis, hypoxia, HIF-1α, tumor progression, Ezrin, L1CAM, vascularization, cellular transdifferentiation, EMT

## Abstract

Hypoxia affects tumor aggressiveness and activates pathways associated with epithelial mesenchymal transition (EMT) which are crucial for tumor progress. In this study, the correlation of hypoxia and EMT with sentinel lymph node status and tumor-specific survival was investigated in primary melanomas. CD34 for capillary count and Hypoxia inducible factor-1α (HIF-1α) as hypoxia indicators as well as Ezrin and L1-Cell Adhesion Molecule (L1CAM), both critical proteins contributing to EMT, were analyzed using immunohistochemistry in 49 melanoma patients with long follow-up (F/U, mean 110 months; range 12–263 months). We found a significant correlation between Breslow tumor thickness and Ezrin expression (*p* = 0.018). L1CAM expression in primary melanoma was significantly associated with HIF-1α expression (*p* < 0.0001) and sentinel lymph node metastasis (*p* = 0.011). Furthermore, low capillary count, reflecting hypoxic condition, was significantly associated with Ezrin expression (*p* = 0.047) and decreased tumor-specific survival (*p* = 0.035). In addition, patients with high Ezrin expression in their primary melanoma had a dramatic loss of life early in their F/U period (mean survival time 29 months; range 15–44 month). Our results highlight the relevance of Ezrin, L1CAM and HIF-1α as prognostic markers in melanoma patients. Additionally, we demonstrate that hypoxia in primary melanoma affects EMT and is at least partly responsible for early metastatic dissemination.

## 1. Introduction

Thanks to new immune modulating and tumor targeting therapies, metastatic melanoma patients have much better tumor free and tumor-specific survival [1,2,3]. However, the incidence of cutaneous melanomas has dramatically increased during the last years, and cutaneous melanomas remain the leading cause of skin cancer death in industrialized countries [4]. It thus becomes all the more important to identify melanoma patients at high risk for metastatic disease. This patient collective could possibly already benefit from initial drug therapy even in the absence of metastases.

Important histopathological prognostic factors on which the TNM stage is already based include nodal status, Breslow tumor thickness, ulceration and mitotic rate [5,6,7]. Additionally, as we have previously shown, the loss of the oncogene p16 and expression of the stem cell marker BMI-1 have an impact on on lymph node status and survival [8,9].

EMT is an important early step in the development of metastases [10,11]. In recent years, a variety of molecular mechanisms that initiate this process have been described. In addition to EMT activation by pathways, epigenetic regulation is also of central importance. Pathways involved in EMT activation include TGF-β, Notch ligand, Wnt, and hypoxia [12,13,14,15,16]. Other proteins involved in EMT include Ezrin and growth factors [17]. Since melanoma originates from neural crest cells, it is more appropriate to speak of tumor cell plasticity.

Melanomas with their stem cell properties have the extreme capacity of plasticity and transdifferentiation [18,19]. We and others have shown that they can even contribute to neo-vascularization [19,20]. Sometimes their melanoma cell origin can just be proofed by molecular analysis or clinical history [21].

During EMT, cell polarization needs to be rearranged from apico-basal to front-rear polarity to enable, migration and subsequent metastasis. In a previous publication, we showed that Ezrin plays an important role in cell polarization and the metastatic process [22].

Ezrin is a protein member of the Ezrin/Radixin/Moesin (ERM) family that are general cross-linkers between cortical actin filaments and plasma cell membranes. Under physiological conditions, Ezrin maintains the actin cytoskeleton, cell morphology and binds to actin filaments, keeping consistent cell contact and mediates signaling pathways to ensure normal shapes of epithelial cells, thereby conserving apico-basal cellular polarity [23]. By reorganizing the actin cytoskeleton, Ezrin also contributes to cell migration and cell division.

L1CAM, a glycoprotein of the immunoglobulin superfamily, is also important for the metastatic process. As its name suggests, L1CAM is involved in cell adhesion and migration [24,25].

Multiple studies implicate the importance of hypoxia, which affects tumor aggressiveness through activating pathways involved in invasiveness and angiogenesis EMT [26,27,28]. In melanoma, we have previously shown an increase of a Tyrosinase Related Protein-2 (TRP-2) negative proliferative subpopulation under hypoxic conditions [29].

The hypoxic response is primarily orchestrated by the HIF signaling pathway. HIFs are heterodimeric transcription factors composed of an α-subunit (of which three isoforms exist, HIF-1α, HIF-2α, and HIF-3α) and a constitutively expressed nuclear β-subunit (of which also three isoform exist, HIF-1β, HIF-2β, and HIF-3β), which maintain oxygen homeostasis by the transcription of genes involved in metabolic remodeling, angiogenic signaling, differentiation, and in cell migration [30]. In the condition of normoxia, HIF-α is unstable. As cellular oxygen levels decrease, HIF-α stabilizes and migrates in the nucleus, where it dimerizes with HIF-β and acts as a transcription factor of several genes involved in angiogenesis, cell survival, EMT, metastization, and drug resistance [31,32]. In fact, it has been suggested that hypoxia contributes to the regulation of every step of metastasis in solid tumors (i.e., from the initial epithelial-mesenchymal transition to the ultimate organotrophic colonization), and therefore positions HIFs as metastatic “master regulators”.

Many of the processes described above are based on studies with cell cultures, animal experiments or neoplasms other than melanoma. Consequently, we were interested in whether hypoxia was relevant for EMT, tumor progress and survival in our melanoma patients collective, consisting of 49 patients with very long follow up (F/U; mean 110 months). As the best indicator for oxygen supply is still vascularization, we analyzed the vessel density using CD34 immunohistochemistry in addition to HIF-1α staining. Furthermore, we performed immunohistochemistry for L1CAM and Ezrin.

Our investigation including important clinical data highlights the importance of hypoxia and EMT in the metastatic process.

## 2. Results

### 2.1. Patients

Forty-nine patients were identified for inclusion in this study. Two patients were missing to F/U immediately upon diagnosis and were excluded from the survival analysis following a sensitivity analysis comparing mean Breslow thickness (2.83 mm) that yielded no significant difference between groups (t-test for Equality of Means t-statistic = 0.043 *p* = 0.9659). 

The patient sample ranged in age between 18 to 75 years at the time of diagnosis (median age = 54.00, mean = 53.53). There were more males than females, 26 (53.2%) had nodular melanomas, 15 of the 49 patients (30.6%) had positive sentinel lymph nodes, the mean Breslow thickness was 2.83 mm (median = 2.20 mm), and 12 patients (25.5%) died of melanoma-related causes before the end of the follow-up period (Table 1).

### 2.2. Relationship between Clinical and Immunohistological Measures of Melanoma

Ezrin expression was absent in 14 of the 49 tumor samples (29%), weak in 25 (51%), and moderate in 10 (20%). Strong expression was never found. Seven patients (14%) had between 10% and 50% positive tumor cells, 16 patients (33%) had between 50% and 90% positive tumor cells, and 12 patients (24%) had over 90% positive tumor cells in their primary melanoma. The Ezrin GIS score ranged between 0 and 8: 14 of 49 patients (29%) scored 0, 14$\;\mathrm{patients}\;\left(29\%\right)\mathrm{scored}\;\leq \;3,\;21\;\mathrm{patients}\;\left(42\%\right)\mathrm{scored}\;>3\;$(Table 2). L1CAM measurements were available in 47 of the 49 tumor samples, all of which demonstrated strong expression. Twenty-two patients of the 47 patients (47%) had between 1–10% positive cells, 11 patients (23.5%) had between 11–50% positive cells, and 11 patients (23.5%) had between 51–90% positive tumor cells (Table 3). Two (4%) tumor samples were negative and one (2%) had L1CAM expression in over 90% of cells. HIF-1α expression was measured in 48 of the 49 samples. HIF-1α GIS scores ranged between 0 and 9: 15 of 48 patients (31%) score 0, 20 $\mathrm{patients}\;\left(42\%\right)\leq \mathrm{score}\;3,\;13\;\mathrm{patients}\;\left(27\%\right)>\mathrm{score}\;3\;$.

CD34 positive capillaries per core was measured in 38 of the 49 patients. Twenty-one of the 38 patients (55%) had <20 CD34 positive capillaries per core, and 17 of the 38 patients (45%) had ≥20 CD34 positive capillaries per core (0.28 mm^2^).

A correlational analysis using Kendall’s tau B showed significantly negative correlation between capillary content and Ezrin GIS score (*T*_b_ = −0.244, *p* = 0.047). There was an important increase in negative correlation if staining intensity was neglected and just the percentage of Ezrin positive cells were correlated (*T*_b_ = −0.410, *p* = 0.001). In addition, there was a positive correlation between Breslow tumor thickness and Ezrin positive cells (*T*_b_ = 0.260, *p* = 0.018). However, this correlation was not present with GIS score calculation. There was no correlation between Ezrin, Ki-67, L1CAM or HIF-1α.

However, we found a significant positive correlation between HIF-1α and L1CAM expression (*T*_b_ = −0.501, *p* = 0.0001).

In addition, the Mann Whitney U test revealed that L1CAM positivity was significantly correlated with sentinel lymph node metastasis (*p* = 0.011). No other markers were significantly correlated with sentinel lymph node metastasis. 

### 2.3. Survival

#### 2.3.1. Kaplan-Meier Method

Twelve patients experienced melanoma-specific death before the end of the study period. Median F/U time of the patients who were living at the point of censoring was 96 months (mean = 110, range = 12–263 months). The five-year survival rate for this sample of melanoma patients was 80%.

Patients with high Ezrin expression in their primary melanoma had obvious poorer overall survival, with a dramatic loss of life early in the F/U period. They had a mean survival time of 29 months (range 15–44 month) compared to 85 months (range 26–150 month) in patients with low Ezrin expression. However, the log rank test was not statistically significant (*p* = 0.120) (Figure 1a and Figure 2a).

Patients who died had significantly lower vascularization (*p* = 0.009). The mean capillary count per core was 10.2 ± SD 8.3 (range 1–25, median 8) compared to 22 ± SD13 (range (2–60, median 20) of those who are still alive. The low CD34 level group, representing hypoxic conditions, had poorer overall survival. All patients except one died within 44 months after diagnosis. However, the log rank test was not statistically significant (*p* = 0.059) (Figure 1b and Figure 2b).

There was no observed difference in survival rates between HIF-1α, L1CAM or Breslow thickness.

As also shown by a multicenter study, our patient collective had poorer overall survival with positive sentinel lymph nodes (log rank Chi Square = 6.230, *p* = 0.013) (Figure 1c) [33]. This supports the quality of our patient cohort. 

Finally, the group with high Ki67 had poorer overall survival, although the log rank test was not statistically significant (*p* = 0.0637) (Figure 1d).

#### 2.3.2. Cox Regression Analysis

Findings from the univariate Cox regression analysis on melanoma-specific mortality appear in Table 4. Sentinel Lymph Node (SLN) positivity was strongly associated with a statistically significant increase in risk of death reflected as a hazard ratio (HR) of 5.338. CD34 capillary count, representing oxygenation, was associated with a statistically significant decreased risk of death reflected as an HR of 0.903. Ki67 was also significantly positively associated with an increased risk of death, where a one percent increase in Ki67 was associated with a 4.6% increase in the risk of death.

CD34 loses its significance as a predictor of survival when adjusted for Ezrin positivity in the multiple regression cox model, but remains close to significance (*p* = 0.05). Ki67 remains a significant predictor of survival when adjusted for Ezrin positivity in the multiple regression model, and CD34 remains a significant predictor of survival when adjusted for Ki67. However, Ki67 loses its predictive ability when adjusted for CD34 (Table 5). 

## 3. Discussion

Our study demonstrates the importance of hypoxia and EMT in melanoma patients’ tumor progress. Low tumor tissue vascularization in primary melanomas, representing ischemia and decreased oxygenation, is significantly associated with worse tumor-specific survival. Additionally, we also found a significant inverse correlation between CD34 positive capillaries and Ezrin expression in tumor tissue. The loss of this CD34 prognostic significance in a multiple regression Cox model with Ezrin underlines its correlation to Ezrin. The huge plasticity of melanoma cells and the ability of transdifferentiation is well known. There are subclones with stem cell properties that enable transdifferentiation, which in epithelial tumors is called EMT [21,34]. Experiments with melanoma cells have shown different gene expression profiles of cells involved in the process of EMT [35,36]. In addition, cells with stem cell properties favor hypoxic conditions [37]. In a previous study we were able to show the prognostic impact of melanoma cells with stem cell properties [19]. This study, with a very long F/U of melanoma patients, underlines the clinical importance of hypoxia in melanoma progress and its most probable relation to EMT.

In previous studies, we reported the importance of Ezrin for melanoma cell polarization plasticity, a process important for cell migration and EMT [22,38]. In addition to rearranging cell polarization from apico-basal to front-rear orientation, the loss of intercellular contact is an equally important factor in EMT. Co-precipitation experiments suggest that Ezrin associates with and regulates E-cadherin and b-catenin, which are critical proteins for intercellular adhesiveness [39,40]. Not surprisingly, increased Ezrin expression indicating dysregulation of cell-cell adhesion is associated with worse prognosis in various epithelial cancers (lung, gastric, breast, ovarian, cervical and hepatic cancer). However, studies with osteosarcoma as well as skin melanoma cell lines and uveal melanoma have shown an association between high Ezrin expression and poor clinical outcome. 

Melanomas and osteosarcomas have no epithelial growth pattern, thus indicating Ezrin’s impact on EMT, specifically tumor cell polarization, adhesion, and migration in those tumor entities. We found just one study analyzing Ezrin expression in melanoma patients. Congruent with this study, we also found a significant correlation between Ezrin expression and Breslow tumor thickness (*p* = 0.018) [41]. In contrast, proliferation did not correlate with Ezrin in our study. This is most probably because all of our patients have a Breslow tumor thickness over 1 mm. 

Inadequate oxygen supply, or hypoxia, is characteristic of the tumor microenvironment, and correlates with poor prognosis and therapeutic resistance. Oxygen levels in tumors depend on several factors, with tumor size and capillary density being two of the most important factors [42]. Low capillary density represents tissue hypoxia, which is a well-known driver of EMT and metastasis through activating a complex machinery of different pathways such as TGF-β Wnt, PI3k/Akt, and Jagged/Notch [26,43]. To our knowledge, this is the first report of a significant inverse correlation between capillary content and Ezrin expression in melanoma tissue. HIF-1α is the main regulator of oxygen homeostasis and regulates over 1000 genes related to low oxygen tension in tissue. These genes are involved in stress adaptation, metabolism, apoptosis, tumor growth, angiogenesis, and invasion [44,45]. However, we found no significant correlation between HIF-1α and Ezrin expression nor with capillary density. This is most likely because HIF-1α is driven by many other factors, as described above, than hypoxia. A previous study also found no significant association between HIF-1α expression in melanoma and outcome [46].

L1CAM belongs to the family of adhesion molecules such as CD44, E-cadherin and N–cadherin, and was first described in neural cell migration [47]. In a previous study, we were able to show its importance in melanoma progression. Knockdown of cellular L1CAM reduced melanoma cell migration and abrogated the chemoresistance against cisplatin [24]. In addition, studies with non-small lung cancer, colorectal and endometrial cancer have demonstrated the importance of EMT and its association with poor clinical outcome [48,49,50].

A recent mouse model of colon cancer demonstrated that L1CAM is upregulated under a hypoxic microenvironment. In our study we also found a significant correlation between L1CAM and HIF-1α (*p* < 0.0001; *T*_b_ = 0.5). However, as discussed above, we found no correlation between HIF-1α and CD34. Interestingly L1CAM expression was significantly correlated with the presence of sentinel lymph node metastasis, which was not found for Ezrin expression. L1CAM as well as Ezrin are proteins involved in EMT and invasion. Perhaps Ezrin has a higher affinity to blood and L1CAM, and a higher affinity for lymphatic vessel invasion. Early blood vessel invasion and dissemination is obvious in patients with high Ezrin expression in their primary melanomas. They show a dramatic loss of life early in the F/U period, with a mean survival time of 29 months.

Here, we neither found a correlation between Ki67 and Ezrin nor a loss of Ki67 predictive character in the multiple Cox regression analysis with Ezrin. These findings indicates that Ezrin expression and Ki67 expression most probably underline different pathways. This agrees with melanoma cell experiments showing that melanoma cells are oscillating between a proliferative and invasive gene signature pattern [51]. In addition, the same authors have shown that a hypoxic environment increases the invasiveness of proliferative melanoma cell cultures. In contrast, invasive phenotype melanoma cells showed no increase in invasive potential upon exposure to hypoxia. Thus, the exposure of proliferative melanoma cells to hypoxic microenvironments is sufficient to increase their invasive potential [52]. This complex interaction between proliferation, invasion and hypoxia could explain how we found that capillary density remains a significant predictor of survival when adjusted for, but a loss of Ki67s prediction when adjusted for capillary density.

The weakness of our study is that the number of melanoma-related deaths is rather low, resulting in a lack of power to detect true associations between predictors and survival. In addition, no transcriptomic or proteomic analyses, which would support our immunohistochemical findings, were performed.

The strength of our study is that the entire patient sample was treated at our University Hospital during a time when tumor targeting therapies or immune modulating therapies were not available. In addition, the F/U is very long.

## 4. Materials and Methods

### 4.1. Patients 

All analyses involving human melanoma tissue were performed in accordance with the ethical committee of Canton Zurich (BASEC-Nr. PB 2017−27). 

The patient collective derives from a tissue microarray (TMA) from a previous melanoma sentinel lymph node study [9,53]. All patients had treatment and F/U at the University Hospital of Zurich according to the Swiss melanoma guidelines [54]. A retrospective chart review of the patient collective was conducted, where clinical and pathological information and clinical outcomes were extracted. The factors included demographics, type of tumor and Breslow tumor thickness. Patients were eligible for this study if they had both a diagnosis of primary cutaneous malignant melanoma between 1999 and 2002 and if they had tumor tissue still available for additional staining. 

### 4.2. Immunohistochemistry

Immunohistochemistry for Ezrin, CD34 and L1CAM was performed on the above-described TMA. Staining was evaluated by two experienced pathologists (U.M. and D.M-P.). Data on immunohistochemistry for Ki67 and HIF-1α were already available from a previous study [29].

CD34 positive capillaries were counted per TMA core representing 0.28 mm^2^. A quantity of less than 20 capillaries per core was considered as low, whereas more than 20 capillaries were considered as high (Figure 3). 

To determine the expression frequencies of Ezrin and HIF-1α a semi quantitative scoring system was applied following the German immunohistochemical scoring (GIS) system in which the final immunoreactive score equaled the product of the percentage of positive cells times the highest staining intensity. The percentage of positive cells was graded as follows: 0: negative; 1: up to 10% positive cells; 2: 11% to 50%; 3: 51% to 90%; and 4: >90%. Staining intensity was graded as follows: 0: negative; 1: weakly positive; 2: moderately positive and 3: strongly positive. For both immunohistochemical stainings, a weak background coloration was observed which was considered as negative. As L1CAM was always strong positive, just the percentage of positive cells was graded as described for Ezrin (Figure 4, Figure 5 and Figure 6).

For Ki67 the hot spot was chosen and positive tumor cells per 100 tumor cells were counted.

Technical information on the antibody used is reported in Table 6.

### 4.3. Statistical Methods

Numerical predictors were visually assessed for normality using normal quintile-quintile (Q-Q) plots. Correlations between numerical predictors with more than two categories were analyzed by calculating Kendall’s tau B correlation coefficient. Comparisons between dichotomous and ordinal variables were tested with the Mann Whitney U test. Comparisons between dichotomous variables were analyzed using a chi-squared test of independence. Survival curves were constructed by the Kaplan-Meier method and compared by the Mantel-Haenszel log-rank test. Kaplan-Meier analyses, numerical and ordinal predictor variables were categorized into dichotomous groups of low and high, based on cut-points easily used in routine practice.

Univariate and multiple regression Cox proportional hazards models were produced to measure the association between predictor variables and survival time. Survival time was defined as the period from the initial melanoma diagnosis to the date of the last follow-up or melanoma-specific death. Patients who were alive at last follow-up or left the study before it concluded (on 29 May 2022) were censored. The prognostic factors included Ezrin GIS score, ki67 percent (proliferation index), CD34 count and HIF-1 α GIS score. Predictor inclusion in the Cox multiple regression survival models was limited to two predictor variables based on the low number of death events and established prior to statistical analysis based on theoretical significance. Patients with missing values for independent variables were excluded from individual analyses rather than using listwise deletion given the small sample size. Model diagnostics were performed, including assessment for non-violation of the assumption of proportional hazards (plots of the survival function and the log minus log survival function by time, and examining interaction terms with time), lack of excess multicollinearity and independence of cases, and indicated no violations of assumptions.

Statistics were considered statistically significant when the *p*-value was less than 0.05. Statistical analysis was performed using SPSS V28.0.1.1 (IBM Corp. Released 2021. IBM SPSS Statistics for Windows, Version 28.0.1.1. Armon, NY, USA: IBM Corp.).

## 5. Conclusions

Ezrin and L1CAM are important prognostic melanoma markers.

Hypoxia in primary melanoma has an important prognostic impact.

Ezrin expression is elevated under hypoxic conditions, is involved in EMT and is at least partly responsible for early metastatic dissemination.

## Figures and Tables

**Figure 1 ijms-23-10745-f001:**
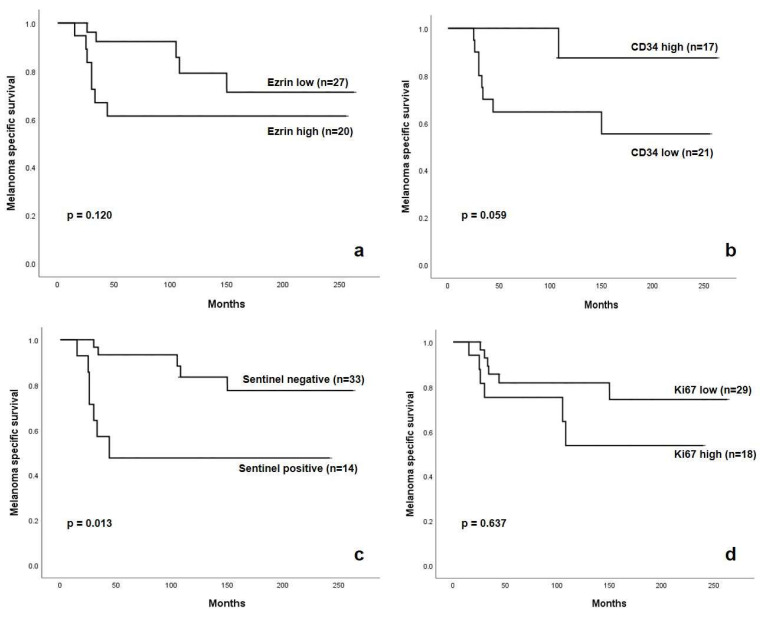
Ezrin expression (**a**), capillary count (CD34; (**b**)), sentinel lymph node status (**c**), Ki-67 (**d**) and tumor specific survival.

**Figure 2 ijms-23-10745-f002:**
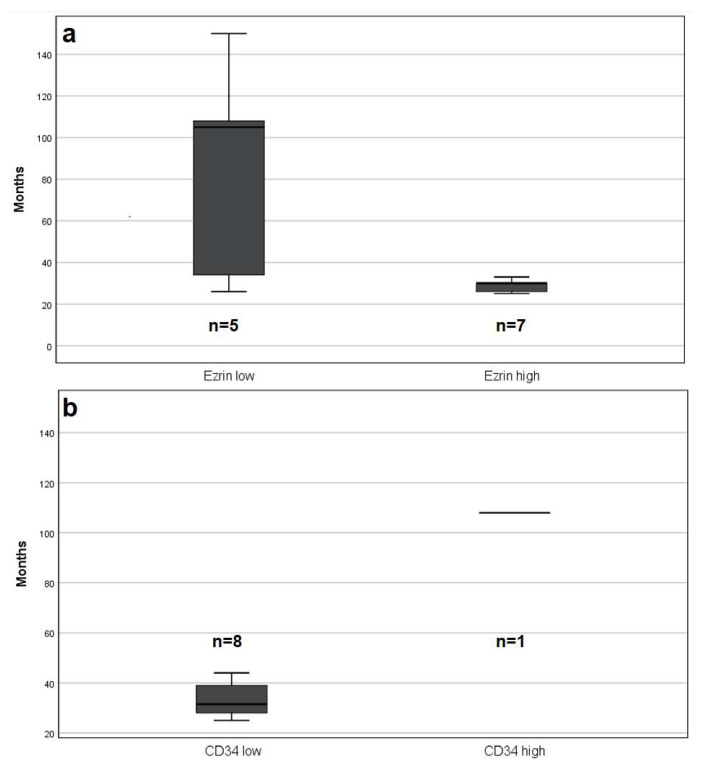
Box-plot graphs illustrating the time differences in months between primary diagnosis and death among patients with low and high Ezrin expression (**a**) as well as in patients with low and high capillary count through CD34 (**b**).

**Figure 3 ijms-23-10745-f003:**
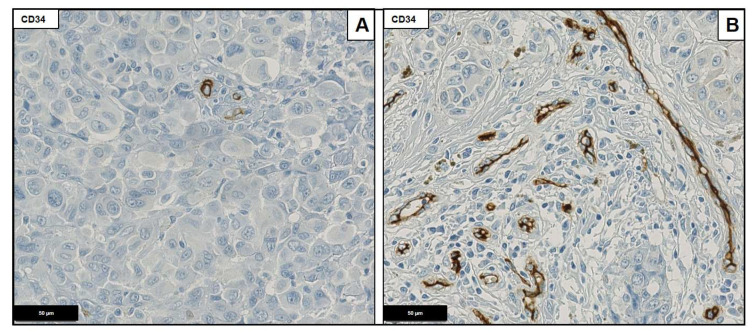
CD 34 staining demonstrating low (<20; **A**) and high (≥20; **B**) capillary density. Magnification ×40.

**Figure 4 ijms-23-10745-f004:**
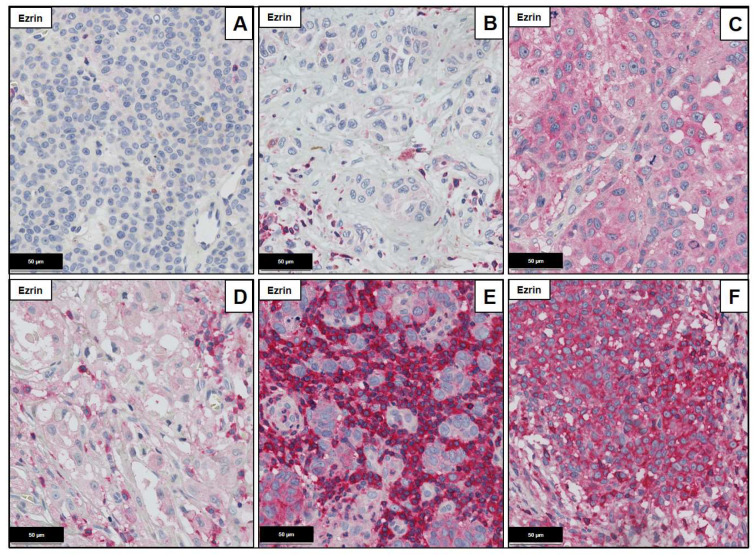
Ezrin staining, magnification 40×. (**A**) Absence of staining (GIS-score 0). (**B**) Weak positivity in <10% (GIS-score 1). (**C**) Weak positivity in 50–90% of tumor cells (GIS-score 3). (**D**) Moderate positivity in <10% of tumor cells (GIS-score 2). (**E**) Moderate positivity in 10–50% of tumor cells (GIS-Score 4). (**F**) Moderate positivity in >90% of tumor cells (GIS-Score 8).

**Figure 5 ijms-23-10745-f005:**
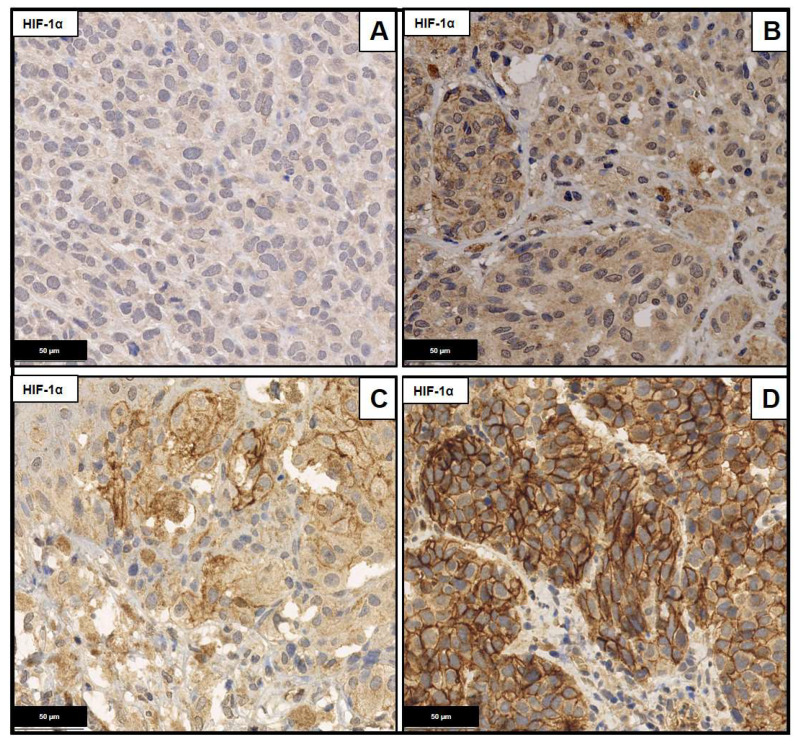
HIF-1α staining, magnification 40×. (**A**) Absence of staining (GIS-score 0). (**B**) Weak positivity in <10% (GIS-score 1). (**C**) Moderate positivity in 10–50% of tumor cells (GIS-Score 4). (**D**) Strong positivity in 50–90% of tumor cells (GIS-Score 8).

**Figure 6 ijms-23-10745-f006:**
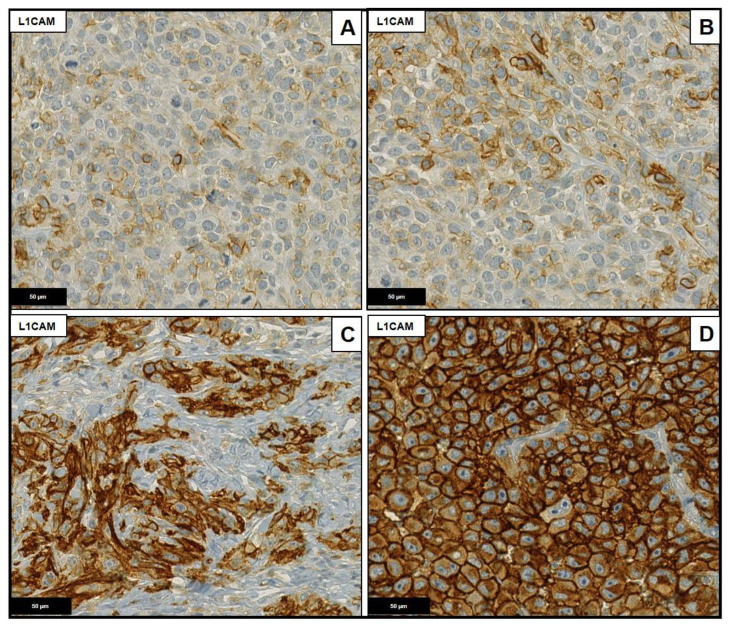
L1CAM staining, magnification 40×. (**A**) Score 1: up to 10% positive cells. (**B**) Score 2: 11% to 50%. (**C**) Score 3: 51% to 90%. (**D**) Score 4: >90%.

**Table 1 ijms-23-10745-t001:** Clinical data.

Variable	Number of Cases (%)
**Age at diagnosis (years)**	
≤65	36 (73.5)
>65	13 (26.5)
**Sex**	
Male	26 (53.1)
Female	23 (46.9)
**Melanoma type**	
Nodular melanoma (NM)	26 (53.0)
Superficial spreading melanoma (SSM)	12 (24.5)
Acral lentiginous melanoma (ALM)	4 (8.2)
Not otherwise specified (NOS)	4 (8.2)
Desmoplastic melanoma	1 (2.0)
Melanoma ex naevo	2 (4.1)
**Sentinel Lymph node**	
Positive	15 (30.6)
Negative	34 (69.4)
**Melanoma-specific death**	12 (24.5)

**Table 2 ijms-23-10745-t002:** Ezrin and HIF-1α expression in primary melanoma.

GIS Score	Ezrin; *n* = 49	HIF-1α; *n* = 48
0	14 (29%)	15 (31%)
≤3	14 (29%)	20 (42%)
>3	21 (42 (%)	13 (27%)

**Table 3 ijms-23-10745-t003:** L1CAM expression in primary melanoma.

% of Positive Melanoma Cells	L1CAM; *n* = 47
0	2 (4%)
1–10	22 (47%)
11–50	11 (23.5%)
51–90	11 (23.5)
>90	1 (2%)

**Table 4 ijms-23-10745-t004:** Predictors of relative excess melanoma-specific mortality from Cox survival models in the melanoma cancer patient sample. * Denotes a significant *p* value of <0.05.

	Unadjusted Hazard Ratio (95% CI)	P (df)
**CD34 (integer, continuous)**	0.917 (0.847–0.994)	**0.035** * (1)
**Hif-1α (GISH)**	1.195 (0.982–1.454)	0.075 (1)
**L1CAM (low vs. high)**	1.165 (0.635–2.140)	0.622 (1)
**Ezrin % positive**	1.346 (0.862–2.101)	0.191
**Ki67 percent (integer, continuous)**	1.046 (1.006–1.087)	**0.022** * (1)
**Sentinel lymph node (negative vs. positive)**	5.338 (1.634–17.446)	**0.006** * (1)
**Breslow thickness (integer, continuous)**	1.134 (0.951–1.351)	0.160 (1)
**Age (integer)**	1.027(0.985–1.072)	0.213
**Sex (male, female)**	1.131 (0.356–3.593)	0.835

**Table 5 ijms-23-10745-t005:** Predictors of melanoma-specific mortality from three multiple regression Cox survival models in the melanoma cancer patient sample. * Denotes a significant *p* value of <0.05.

	Unadjusted Hazard Ratio (95% CI)	P (df)	Adjusted Hazard Ratio (95% CI)	P (df)
Ezrin % positive	1.346 (0.862–2.101)	0.191	1.423 (0.682–2.967)	0.347(1)
CD34	0.917 (0.847–0.994)	0.035 * (1)	0.932(0.856–1.014)	0.102 (1)
Ezrin (%positive)	1.346 (0.862–2.101)	0.191	1.381 (0.873–2.184)	0.168(1)
Ki67 percent (integer, continuous)	1.046 (1.006–1.087)	0.022 * (1)	1.047(1.007–1.089)	0.020 * (1)
Ki67 percent (integer, continuous)	1.046 (1.006–1.087)	0.022 * (1)	1.043 (0.991–1.098)	0.110(1)
CD34	0.917 (0.847–0.994)	0.035 * (1)	0.911(0.839–0.839)	0.025 * (1)

**Table 6 ijms-23-10745-t006:** Details of immunohistochemical antibodies.

Antibody	Clone	Dilution	Source	Platform
**CD34**	QBEnd/10	1:800	Serotec Ltd.	Ventana
**L1CAM**	EPR18750	1:500	Abcam limited	Ventana
**HIF-1α**	Mgc3	1:400	Abcam limited	Ventana
**Ezrin**	polyclonal	1:100	Cell Signaling	Bond MAX

## Data Availability

Study relevant data are anonymized available upon justified request addressed to the corresponding author.

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
