# Peer review of "Hypoxia and Ezrin Expression in Primary Melanoma Have High Prognostic Relevance"

_ijms, 2022, doi:10.3390/ijms231810745_

Round 1

Reviewer 1 Report

The manuscript entitled “Hypoxia and Ezrin expression in primary melanoma have high prognostic relevance” by Maccio U. et al., was presented as an original Article in which the authors analyzed the correlation of hypoxia and epithelial mesenchymal transition with sentinel lymph node status and tumor-specific survival in primary melanomas and highlighted the relevance of Ezrin, L1CAM and HIF-1α as prognostic markers in melanoma. The introduction needs to be revised, adding more detailed novelty about this study. Although the conclusions are in line with the presented data, the results could be improved including several experimental approaches and different methodologies. Therefore, the manuscript could be of potential interest in the field of cancer, providing interesting new insights especially in melanoma research, but several major points need to be addressed before the manuscript may be formally accepted for publication:

A)  The authors extensively described the relationship between clinical and immunohistological measurements of melanoma samples in the paragraph 2.2. However, to facilitate and make more immediate the interpretation of the data, the authors should: I) schematize them graphically, generating a Venn diagram which shows all possible relationships between Ezrin/HIF-1α/CD34/L1CAM expression and II) show a scatterplot for each correlational analysis. Furthermore, it would be appropriate to provide a table showing clinical and immunohistological data for each patient enrolled in the clinical study.

B)   Although the authors performed immunohistochemical analyses on a consistent number of human melanoma tissue samples, they showed only a representative staining for each antibody tested. Please, extend the data in Figure 2, showing a panel with at least 10 sections from different patients. Figure 2 needs to be improved. The reviewer suggests: I) to indicate next to each sections the name of the antibody they are stained for and II) to show “LOW EXPRESSION” and “HIGH EXPRESSION” at the top of the two columns of immunohistological sections. Finally, most importantly, scale bars are missing; please add them and specify the magnification (x) and the length (μm) in the Figure Legend. Also Figure Legend 2 needs to be correct: the authors did not cite panel D. 

C)  In order to define the prognostic role of Ezrin, L1CAM and HIF-1α in melanoma patients, further analyses would be required. To this aim, immunohistochemical analyses could be supported with molecular data, evaluating mRNA and/or protein expression levels in the same tissue samples. 

D)  The authors concluded that Ezrin expression is involved in epithelial mesenchymal transition, and it is partly responsible for early metastatic dissemination. To support this conclusion, they must corroborate the presented data evaluating i.e. E- to N-Cadherin switch and the expression levels of key regulators of EMT pathway.

E) Please, double check and correct the typos of words/acronyms/abbreviations/tenses in the text. The reviewer suggests: 

I) to capitalize the first letters of full words that define the acronyms/abbreviations (i.e. Epithelial Mesenchymal Transition, EMT, L1 Cell Adhesion Molecule, L1CAM, Hypoxia-Inducible Factor, HIF

II)      to define all abbreviations before introducing them in the text. Please, specify the extended form of each acronym/abbreviation (i.e. Tumor, Node, Metastasis, TNM, German Immunohistochemical Scoring, GIS; Sentinel Lymph Node, SLN)

III)    to add the acronym in parentheses after the written-out form (i.e. page 2, line 66: Hypoxia-Inducible Factor, HIF

IV)  to introduce the acronyms for the terms that are repeated two or more times in the text. Once you have entered an acronym, make sure that you no longer repeat the extended form in the text (i.e. page 2, lines 63-64/line 76-77: Epithelial Mesenchymal Transition, EMT).

Author Response

  1. The authors extensively described the relationship between clinical and immunohistological measurements of melanoma samples in the paragraph 2.2. However, to facilitate and make more immediate the interpretation of the data, the authors should: I) schematize them graphically, generating a Venn diagram which shows all possible relationships between Ezrin/HIF1α/CD34/L1CAM expression and II) show a scatterplot for each correlational analysis. Furthermore, it would be appropriate to provide a table showing clinical and immunohistological data for each patient enrolled in the clinical study.

We have made a new box plot graph, which illustrates that patients with low vascularization or high ezrin expression have a much shorter survival (figure 2).

A table containing all 49 patients is confusing and does not provide any additional information on the already existing clinical table 1. However, we added table 2, which summarizes the immunohistochemical results.

 B. Although the authors performed immunohistochemical analyses on a consistent number of human melanoma tissue samples, they showed only a representative staining for each antibody tested. Please, extend the data in Figure 2, showing a panel with at least 10 sections from different patients. Figure 2 needs to be improved. The reviewer suggests: I) to indicate next to each sections the name of the antibody they are stained for and II) to show “LOW EXPRESSION” and “HIGH EXPRESSION” at the top of the two columns of immunohistological sections. Finally, most importantly, scale bars are missing; please add them and specify the magnification (x) and the length (μm) in the Figure Legend. Also Figure Legend 2 needs to be correct: the authors did not cite panel D.

We extended the data in figure 2 and show more representative samples of different patients. The scale bars are of better quality. Name of the antibody is inside the figure. In figure legend 2 panel D is now correctly cited.

C. In order to define the prognostic role of Ezrin, L1CAM and HIF-1α in melanoma patients, further analyses would be required. To this aim, immunohistochemical analyses could be supported with molecular data, evaluating mRNA and/or protein expression levels in the same tissue samples.

This is an interesting point and would be appropriate to examine these data in another study. Since in the present work we have focused our attention to immunohistochemistry, we have added the lacking of molecular analysis as a possible limitation of the study.

D. The authors concluded that Ezrin expression is involved in epithelial mesenchymal transition, and it is partly responsible for early metastatic dissemination. To support this conclusion, they must corroborate the presented data evaluating i.e. E- to N-Cadherin switch and the expression levels of key regulators of EMT pathway.

As described in the introduction we are aware that EMT is a highly complex process involving many pathways and molecules. In this study we focused on Ezrin and L1CAM. Beside many other candidates E- to N-Cadherin switch would be highly interesting to analyze in a further study with our patient collective.

E. Please, double check and correct the typos of words/acronyms/abbreviations/tenses in the text. The reviewer suggests: I) to capitalize the first letters of full words that define the acronyms/abbreviations (i.e. Epithelial Mesenchymal Transition, EMT, L1 Cell Adhesion Molecule, L1CAM, HypoxiaInducible Factor, HIF) II) to define all abbreviations before introducing them in the text. Please, specify the extended form of each acronym/abbreviation (i.e. Tumor, Node, Metastasis, TNM, German Immunohistochemical Scoring, GIS; Sentinel Lymph Node, SLN) 2 III) to add the acronym in parentheses after the written-out form (i.e. page 2, line 66: HypoxiaInducible Factor, HIF) IV) to introduce the acronyms for the terms that are repeated two or more times in the text. Once you have entered an acronym, make sure that you no longer repeat the extended form in the text (i.e. page 2, lines 63-64/line 76-77: Epithelial Mesenchymal Transition, EMT).

We have checked the entire manuscript again and corrected these critical points according to the suggestions.

Reviewer 2 Report

In this work by Umberto Maccio et al entitled “Hypoxia and Ezrin expression in primary melanoma have high 2 prognostic relevance”, the authors aim to determine correlations between expression of ezrin, L1CAM and HIF-1 and melanoma diagnostics, prognosis and general outcomes using numerous patients. Expression of the different factors were carried out using immunohistochemistry and signal intensities correlated to patient information.

Their work shows that there are similarities between ezrin, CD34 and L1CAM expression and the severity of melanoma and that HIF-1a also offers some analysis where circumstances support a potential role in carcinogenesis.

The work is of interest and offers some data which can be considered to be supportive. There is however no direct links between the different factors tested and it seems much of the work is presented at random. If anything, the authors should try to link most if not all the different factors to validate the logics of their analysis.

The same can be said for the analysis where sometimes some are presented, but others are lacking e.g. where is the data for L1CAM in Figure 1 or table 3?

Other points to consider:

The authors keep on referring to the importance of EMT in melanoma progression but fail to provide any work highlighting the EMT process in their studies. Factors such as ezrin have of course been linked to EMT but providing coincidental levels of their differential expression does not validate the EMT process at least not sufficiently to keep on referring to it with such importance throughout.

Author Response

In this work by Umberto Maccio et al entitled “Hypoxia and Ezrin expression in primary melanoma have high 2 prognostic relevance”, the authors aim to determine correlations between expression of ezrin, L1CAM and HIF-1 and melanoma diagnostics, prognosis and general outcomes using numerous patients. Expression of the different factors were carried out using immunohistochemistry and signal intensities correlated to patient information.

Their work shows that there are similarities between ezrin, CD34 and L1CAM expression and the severity of melanoma and that HIF-1a also offers some analysis where circumstances support a potential role in carcinogenesis.

The work is of interest and offers some data which can be considered to be supportive. There is however no direct links between the different factors tested and it seems much of the work is presented at random. If anything, the authors should try to link most if not all the different factors to validate the logics of their analysis.

Our study has been carefully conducted and shows clearly statistically proven results. Our careful approach is well documented by being able to reproduce already proven prognostic markers such as sentinel lymph node status and proliferation in addition to new findings.

The same can be said for the analysis where sometimes some are presented, but others are lacking e.g. where is the data for L1CAM in Figure 1 or table 3?

L1CAM was already shown in figure 1, but we have improved this figure according to the points suggested by Reviewer 1. In table 3 multivariate analysis are reported. Since the results of L1CAM were not significant in the univariate analysis, it does not make sense to analyze L1CAM in the multivariate analysis.

Other points to consider:

The authors keep on referring to the importance of EMT in melanoma progression but fail to provide any work highlighting the EMT process in their studies. Factors such as ezrin have of course been linked to EMT but providing coincidental levels of their differential expression does not validate the EMT process at least not sufficiently to keep on referring to it with such importance throughout.

As already citied in our introduction and discussion, already basic researches have shown previously in journals with high impact that Ezrin is involved in EMT. Therefore, our findings should not be regarded as coincidental. We added reference 17 another study illustrating the importance of Ezrin on TMT

Reference 17: Fröse J, Chen MB, Hebron KE, Reinhardt F, Hajal C, Zijlstra A, Kamm RD, Weinberg RA. Epithelial-Mesenchymal Transition Induces Podocalyxin to Promote Extravasation via Ezrin Signaling. Cell Rep. 2018 Jul 24;24(4):962-972. doi: 10.1016/j.celrep.2018.06.092. PMID: 30044991; PMCID: PMC6181240.

Round 2

Reviewer 1 Report

POINT A, REVIEWER RESPONSE: It would be appropriate to merge and reorganize Figure 1 and Figure 2, in order that any figure/panel can be cited in correct order through the text. 

POINT E, REVIEWER RESPONSE: Make the following text corrections:

1)   List of Abbreviations: capitalize the first letters of full words that define the acronyms/abbreviations in EMT = Epithelial Mesenchymal Transition, HIF = Hypoxia Inducible Factor, F/U = Follow-Up

2)   Introduction: although already introduced in the abstract, please, specify the extended form of each acronym/abbreviation also in the Introduction section

-       Page 2, Line 54: Epithelial Mesenchymal Transition (EMT)

-       Page 2, Line 75: L1 Cell Adhesion Molecule (L1CAM)

-    Page 2, Line 83: HIF (Hypoxia-Inducible Factor)
